# Intracranial Pressure Patterns and Neurological Outcomes in Out-of-Hospital Cardiac Arrest Survivors after Targeted Temperature Management: A Retrospective Observational Study

**DOI:** 10.3390/jcm10235697

**Published:** 2021-12-03

**Authors:** Hogul Song, Changshin Kang, Jungsoo Park, Yeonho You, Yongnam In, Jinhong Min, Wonjoon Jeong, Yongchul Cho, Hongjoon Ahn, Dongil Kim

**Affiliations:** 1Department of Emergency Medicine, Chungnam National University Hospital, Daejoen 35015, Korea; songhg@cnuh.co.kr (H.S.); changsiny@naver.com (C.K.); yyh1003@hanmail.net (Y.Y.); gardenjun@hanmail.net (W.J.); boxter73@naver.com (Y.C.); jooniahn@daum.net (H.A.); 2Department of Emergency Medicine, College of Medicine, Chungnam National University, Daejoen 35015, Korea; ynsoft@naver.com (Y.I.); laphir2006@naver.com (J.M.); 3Department of Emergency Medicine, Sejong Chungnam National University Hospital, Sejong 30099, Korea; 4Department of Computer Science and Engineering, Chungnam National University, Daejoen 35015, Korea; dkim@cnu.ac.kr

**Keywords:** out-of-hospital cardiac arrest, intracranial pressure, blood–brain barrier, prognosis, cerebral edema, target temperature management

## Abstract

We aimed to investigate intracranial pressure (ICP) changes over time and the neurologic prognosis for out-of-hospital cardiac arrest (OHCA) survivors who received targeted temperature management (TTM). ICP was measured immediately after return of spontaneous circulation (ROSC) (day 1), then at 24 h (day 2), 48 h (day 3), and 72 h (day 4), through connecting a lumbar drain catheter to a manometer or a LiquoGuard machine. Neurological outcomes were determined at 3 months after ROSC, and a poor neurological outcome was defined as Cerebral Performance Category 3–5. Of the 91 patients in this study (males, *n* = 67, 74%), 51 (56%) had poor neurological outcomes. ICP was significantly higher in the poor outcome group at each time point except day 4. ICP elevation was highest between days 2 and 3 in the good outcome group, and between days 1 and 2 in the poor outcome group. However, there was no difference in total ICP elevation between the poor and good outcome groups (3.0 vs. 3.1; *p* = 0.476). All OHCA survivors who had received TTM had elevated ICP, regardless of neurologic prognosis. However, the changing pattern of ICP levels differed depending on the neurological outcome.

## 1. Introduction

Hypoxic-ischemic brain injury (HIBI), which often occurs after cardiac arrest (CA), increases the permeability of the blood–brain barrier’s (BBB) tight junctions due to microvascular damage induced through oxidative stress, which leads to BBB disruption and vasogenic edema and causes an elevation in intracranial pressure (ICP) [1,2,3]. Increased ICP is a serious post-CA complication closely associated with poor neurological outcomes [4,5,6]. Therefore, some studies have reported that ICP control may play an important role in improving mortality and neurological prognosis in post-CA care [7,8,9]. However, recently published international guidelines for post-CA care do not provide guidance concerning appropriate ICP levels and methods for reducing ICP [10].

ICP is mostly measured in CA survivors through indirect methods, such as optic nerve sheath diameter (ONSD) measurements using ultrasound, computed tomography (CT), magnetic resonance imaging (MRI), or non-invasive methods using transcranial Doppler (TCD) [7,8,11]. Although these methods are commonly used, their accuracy remains questionable. A recent study on the association between ICP and ONSD, TCD, and jugular venous bulb pressure reported the strongest correlation between ONSD and intracranial hypertension [12]. However, the correlation (*R* = 0.53) was low, and most patients who underwent targeted temperature management (TTM) had an ICP < 20 mmHg. This raises many questions concerning the usefulness of these methods in clinical practice [5,8,13,14]. Few studies have measured ICP directly, most with small sample sizes of <50 cases [12,14,15]. Among these studies, one study reported that ICP rises during the rewarming stage after the maintenance stage in TTM; however, no mention of differences in ICP elevation relating to neurological outcomes was reported [15]. Changes in ICP are presumed to have significant variability over time and differing patient neurological outcomes. ICP measurement through lumbar puncture (LP) is known to be less invasive than other direct measurement methods and shows a high correlation of 0.98 with directly measured ICP [16].

We hypothesized that patients with poor neurological outcomes would have higher ICP levels than those with good neurological outcomes and that the ICP elevation in both groups would be minimal during the TTM maintenance stage. We aimed to investigate the association between ICP levels and neurological outcomes for 3 days after the return of spontaneous circulation (ROSC) in patients with out-of-hospital CA (OHCA) treated with TTM.

## 2. Materials and Methods

### 2.1. Study Design and Population

In this retrospective observational study, we used prospectively collected data from adult comatose OHCA survivors treated with TTM at the Chungnam National University Hospital (CNUH) in Daejeon, Korea, between May 2018 and December 2020. Of the 91 patients in this study, 37 were also included in two other ONSD studies conducted at CNUH [7,8]. The Institutional Review Board of CNUH approved this study (CNUH-2021-05-023) and waived the need for informed patient consent, because of the study’s retrospective nature.

We included adult patients ≥18 years of age who were treated with TTM after cardiac arrest and whose ICP levels were measured. We excluded patients who had experienced a traumatic cardiac arrest or an interrupted TTM owing to transfer, those who were ineligible for LP (i.e., those with brain CTs showing severe cerebral edema, those treated with antiplatelet and anticoagulation therapy, or those with coagulopathy), those who were receiving extracorporeal membrane oxygenation, and those who had no next of kin to provide consent for an LP procedure.

### 2.2. TTM and Sedatives

During the study period, all comatose OHCA survivors were considered eligible for TTM. A target temperature of 33 °C was maintained for 24 h using an Arctic Sun^®^ (Energy Transfer Pads™; Medivance Corp., Louisville, CO, USA) feedback-controlled surface cooling device. Upon completion of the TTM maintenance period, the patients were rewarmed to 37 °C at a rate of 0.25 °C/h, and the temperatures were monitored using a bladder temperature probe (Figure 1).

All patients received sedatives and a neuromuscular blocking agent (NMBA) during TTM. Midazolam (0.05 mg/kg intravenous bolus followed by a titrated intravenous continuous infusion between 0.05 and 0.2 mg/kg/h) and cisatracurium (Nimbex) (0.15 mg/kg intravenous bolus followed by an infusion up to 5 μg/kg/min) were administered for patient sedation and to control shivering, respectively.

### 2.3. Data Collection

We obtained data on the following parameters from the hospital records: age; sex; pre-existing illness; witnessed collapse; bystander cardiopulmonary resuscitation (CPR); first monitored rhythm; etiology of CA; time from collapse to CPR (no flow time); time from CPR to ROSC (low flow time); time from ROSC to obtaining ICP via LP; serum pH, lactate, albumin, creatinine, troponin I, and procalcitonin levels immediately after ROSC; albumin quotient at 0, 24, 48, and 72 h after ROSC; and the sequential organ failure assessment (SOFA) score within the first 24 h after admission. We analyzed PaO_2_, PaCO_2_, and mean arterial pressure (MAP) data collected for the first 24 h after ROSC, and NMBA and sedative agent data were collected for 72 h using a time-weighted average (TWA) [16,17]. Intracranial hypertension was determined as ICP > 20 mmHg. We assessed BBB disruption using the cerebrospinal fluid (CSF)–serum albumin quotient (Q_A_), which is widely accepted as the gold standard for the functional assessment of BBB disruption [17]. We assessed neurological outcomes at 3 months after ROSC, evaluated according to the Glasgow–Pittsburgh Cerebral Performance Category (CPC) scale. We dichotomized results into good neurological (CPC 1–2) and poor neurological (CPC 3–5) outcomes.

### 2.4. Measurement of ICP and Q_A_

An 18-gauge lumbar catheter placement was performed using a Hermetic^TM^ lumbar catheter accessory kit (Integra Neurosciences, Plainsboro, NJ, USA); the patient was lying in a lateral decubitus position with the hips and knees flexed. ICP was measured using a manometer (between April 2018 and December 2018) and a LiquoGuard^®^ pump system (Möller-Medical, Fulda, Germany) (between January 2019 and December 2020) and recorded on days 1, 2, 3, and 4. In this study, the ICP was measured immediately after ROSC on day 1, at the end of the 33 °C maintenance period on day 2, 24 h after day 2 on day 3, and after 48 h after day 2 on day 4. Day min and day max were determined as the minimum and maximum ICP levels on days 1–4. CSF albumin and serum albumin samples were simultaneously obtained at the time of ICP measurements on days 1–4. The Q_A_ was calculated using the following formula: Q_A_ = (albumin_(CSF)_)/(albumin_(serum]_) [9,18].

### 2.5. Statistical Analysis

Categorical variables are presented as frequencies with percentiles, and continuous variables are presented as median values with interquartile ranges because all continuous variables had a non-normal distribution. We compared categorical variables between the groups using χ^2^ tests with a continuity correction in 2 × 2 tables or Fisher’s exact test, as appropriate. We compared continuous variables between the groups using Mann–Whitney *U*-tests. A Friedman test with a Wilcoxon-signed rank-sum test after Bonferroni correction was used to evaluate the changes in ICP levels over time in each neurological outcome group. Additionally, the longitudinal ICP data were analyzed using the generalized estimating equation (GEE) and Bonferroni’s post hoc test. To assess the predictive accuracy for a 3-month poor neurological outcome, receiver operating characteristic (ROC) curves were constructed for the prognostic values of ICP on days 1, 2, 3, and 4 and at the maximum points. The optimal cut-off values were determined using Youden’s index (sensitivity + specificity − 1). Statistical analyses were performed using SPSS version 24 software (IBM Corp., Armonk, NY, USA) and the MedCalc program, version 15.2.2 (MedCalc Software, Mariakerke, Belgium). The significance level was set to *p* < 0.05.

## 3. Results

### 3.1. Characteristics of the Study Population

A total of 115 OHCA comatose survivors were treated with TTM during the study period. Of these patients, 6 received ECMO, 14 were ineligible for lumbar catheter placement, and 4 had a non-functional lumbar catheter due to obstruction. Finally, 91 patients were included in the study. At 3 months after ROSC, 40 (44.0%) patients were assigned to the good neurological outcome group, while 51 (56.0%) were assigned to the poor neurological outcome group, as shown in Figure 2. The demographic and OHCA characteristics, stratified according to neurological outcomes at 3 months, are shown in Table 1. The good neurological outcome group had higher rates of witnessed events, bystander CPR, shockable rhythm, and cardiac etiology, as well as shorter no-and low-flow times, than the poor neurological outcome group. Additionally, the good neurological outcome group had lower troponin I levels and SOFA scores and higher albumin levels, TWA values for MAP, and NMBA requirements than those with poor neurological outcomes. In this study, four adverse events occurred during lumbar drains, namely, reversible, small-scale subdural hematoma (*n* = 1); pneumocephalus (*n* = 2); and catheter fracture (*n* = 1). The catheter fracture was surgically removed 12 days after completion of the TTM (Figure 3).

### 3.2. The Relationship between ICP and Neurological Outcome Groups

On days 1–4, day min and day max ICP levels between the good and poor neurological outcome groups were: 10.4 mmHg (8.0–12.4) vs. 12.5 mmHg (10.0–15.4), (*p* = 0.006); 10.7 mmHg (9.7–13.8) vs. 16.0 mmHg (13.1–17.8), (*p* < 0.001); 14.0 mmHg (11.3–15.0) vs. 16.0 mmHg (13.7–18.5), (*p* = 0.002); 14.0 mmHg (11.0–15.0) vs. 15.0 mmHg (12.2–16.9), (*p* = 0.088); 9.9 mmHg (7.4–11.0) vs. 11.4 mmHg (9.5–14.0), (*p* = 0.002); and 14.8 mmHg (12.7–16.6) vs. 17.1 mmHg (15.3–21.3), (*p* = 0.001), respectively (Figure 4). Moreover, peak ICP levels during days 1–4 were also higher in the poor neurological outcome group than in the good neurological group (17.0 mmHg (15.3–21.1) vs. 14.8 mmHg (12.7–16.2), respectively; *p* = 0.002).

Table 2 shows the ICP levels for post-cardiac arrest care over time (for 72 h), and the results of the GEE statistical analysis with a post hoc test using the Bonferroni procedure. ICP in good and poor neurological outcomes was increased significantly over time in both groups (*P*_GEE_ < 0.001) (Table 2 and Figure 4). However, the discrepancy in the increasing pattern of ICP between groups was observed on the post hoc analysis. ICP was significantly increased in the good neurological outcome group between days 2 and 3 (*p*_GEE_ = 0.001), whereas the poor neurological outcome group showed significant increase of ICP between days 1 and 2 (*P*_GEE_ = 0.001). The other periods in both groups did not show a significant increase of ICP.

Using ROC analyses, the optimal cut-off values of ICP for prediction of poor neurological outcomes at 3 months after ROSC were determined as follows: day 1: >11.8 mmHg (sensitivity, 56.9%; specificity, 70.0%); day 2: >14.0 mmHg (sensitivity, 66.7%; specificity, 83.3%); day 3: >15 mmHg (sensitivity, 61.9%; specificity, 74.3%); day 4: >14.8 mmHg (sensitivity, 61.1%; specificity, 65.7%); day min: >11.0 mmHg (sensitivity, 55.6%; specificity, 85.3%); and day max: >15.0 mmHg (sensitivity, 75.0%; specificity, 65.7%; Figure 5).

### 3.3. The Relationship between Q_A_ and Neurological Outcome Groups

On days 1–4, the poor neurological outcome group showed significantly higher Q_A_ values than the other group at all time points. However, in both groups, the Q_A_ values were the highest on the second day, with a gradually decreasing pattern (Table 3).

## 4. Discussion

In this study, we aimed to investigate ICP changes over time and neurologic prognosis in OHCA survivors who received TTM. We observed that all OHCA survivors who received TTM showed elevated ICP regardless of the neurologic prognosis, and there were no differences in the levels of ICP elevation. ICP levels were significantly higher in the poor neurological outcome group than in the good neurological outcome group at each time point, except at day 4. Additionally, the degree of elevation in ICP levels differed between the two groups, with the highest degree of elevation between days 2 and 3 in the good neurological outcome group, and between days 1 and 2 in the poor neurological outcome group. However, ICP was within the normal range at each time point in both groups.

Few clinical studies have defined the degree of inhibition of therapeutic hypothermia (TH) that affects ICP levels in post-CA care. Naito et al. investigated a cohort of nine patients with cardiac arrest who underwent mild TH and reported that ICP increased from 6 mmHg to 14 mmHg during hypothermia (34 °C) and increased to 16 mmHg after rewarming was completed. The incremental increase in ICP was noted to be higher in the poor outcome group [15]. Sekhon et al., reported that intracranial hypertension only occurred in 279 (11%) of 2419 data points with significant heterogeneous ICP patterns apparent over time between individual patients [5]. Importantly, both studies were limited to small sample sizes and had inadequate information concerning the association between ICP changes and neurological outcomes. These studies did not provide a detailed explanation regarding changes in ICP levels. A reinterpretation of our findings may provide a better understanding of their results. In Naito et al.’s study, a higher elevation of ICP was observed during the hypothermia period than during the rewarming period. Of a total cohort of 9 patients, 3 had good outcomes (CPC 1–2), and 6 had poor outcomes (CPC 3–5) [15]. In our study, the ICP elevation was highest in the poor outcome group during maintenance hypothermia (33 °C) and in the good outcome group during rewarming. Our results are similar to those of Naito et al. in terms of the ICP changes observed in the poor outcome group. Considering the findings of Sekhon et al., our study also showed different patterns of ICP changes depending on the prognosis and measurement time [5]. Notably, Sekhon et al. considered different monitoring times; specifically, the start time and duration of ICP monitoring varied from 6–43 h and 8–94 h after ROSC, respectively, which may explain why a distinctive pattern in terms of ICP changes was not identified.

Appropriate treatment times and methods to achieve optimal ICP levels and control ICP in post-CA care need to be clarified. In our previous study using CSF pressure in 83 patients post-CA, the predictive threshold for poor neurological outcomes immediately after ROSC was 14.7 mmHg [6]. Similarly, several studies have reported that only a small number of patients with intracranial hypertension were observed in the poor outcome group [5,14,19]. While ICP elevation can be observed in most patients with OHCA, both poor and good outcome groups may have cut-off values in the normal ICP range. In our study, ICP levels predicting poor neurological outcomes were 11.8 mmHg, 14.0 mmHg, 15.0 mmHg, and 14.8 mmHg on days 1, 2, 3, and 4, respectively, which were all within the normal range. Recently published international guidelines do not indicate appropriate ICP levels in post-CA care. However, the ICP cut-off levels predicting poor neurological outcomes in this study are not clinically relevant, due to their low sensitivity and specificity; thus, further research is required in the future. The current recommendation only suggests positioning the patient with a 30° head elevation to decrease ICP [10]. From a clinical perspective, hyperosmolar therapy such as mannitol and hypertonic saline is administered to reduce ICP that is secondary to brain edema by reducing swelling through draining fluid from the brain [20]. However, if the BBB is disrupted, osmotically active particles can accumulate inside the brain and aggravate edema formation [20,21]. In our study, the Q_A_ values of the poor prognosis group on days 1, 2, 3, and 4 were 0.01, 0.03, 0.029, and 0.025, respectively, reflecting severe BBB disruption (Q_A_ > 0.02), except immediately after ROSC. Based on these results, if hyperosmotic therapy is performed for ICP control, good effects can be expected only if hyperosmotic therapy is performed as soon as possible, that is, immediately after ROSC to the end of the TH maintenance stage. After the rewarming period or during the late TH maintenance period, mechanical methods to control ICP by draining CSF fluid through a lumbar catheter may be more effective than hyperosmolar therapy for ICP control [22,23]. These suggested methods for ICP control need to be verified through detailed clinical studies in the future.

This study had several limitations. First, this was a single-center retrospective study with a small number of patients; therefore, a multicenter prospective study is recommended to enhance the generalizability of the findings. Second, of 115 patients who underwent TTM during this study period, 14 (12.2 %) were excluded because lumbar drains were not inserted. Lumbar drains in post-CA care are rare in clinical practice and are complex to apply. This exclusion has caused selection bias and could limit the generalizability of our findings. Third, ICP levels were measured during the study period using a manometer and a LiquoGuard. While a LiquoGuard can measure data every 6–10 s continuously, due to the limitations of this retrospective study, the relevant records could not be obtained, and only the ICP levels recorded at 24 h intervals in the medical records were evaluated. Fourth, investigator bias might have been unavoidable since the treating physicians were exposed to the results of the ICP, Q_A_, and serum neuron-specific enolase levels.

## 5. Conclusions

Among OHCA survivors who received TH, patients with poor neurological outcomes showed higher ICP levels than those with good neurological outcomes. There was no difference in ICP levels immediately after ROSC and 72 h later between the two groups. A different pattern was observed; the group with the poor neurological prognosis had the highest ICP levels during TH, while the group with a good neurological prognosis had the highest ICP levels during the rewarming period. However, ICP was within the normal range at each time point in both groups.

## Figures and Tables

**Figure 1 jcm-10-05697-f001:**
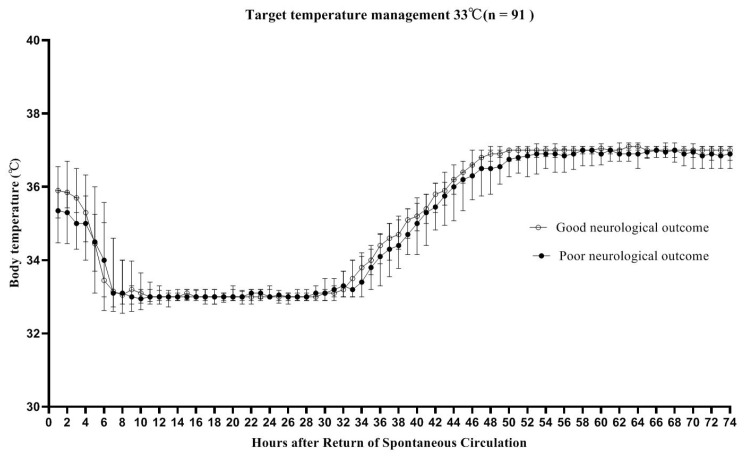
Target temperature management, 33 °C (*n* = 91). The ┬ bars indicate the interquartile ranges of the good and poor outcome groups. The target temperature of both groups was 33 °C (32–34 °C), and the duration of cooling was 24 h. Upon completion of the targeted temperature management (TTM) cooling period, the patients were rewarmed to 37 °C at a rate of 0.25 °C/h. Only patients with recorded temperatures were included in the analysis.

**Figure 2 jcm-10-05697-f002:**
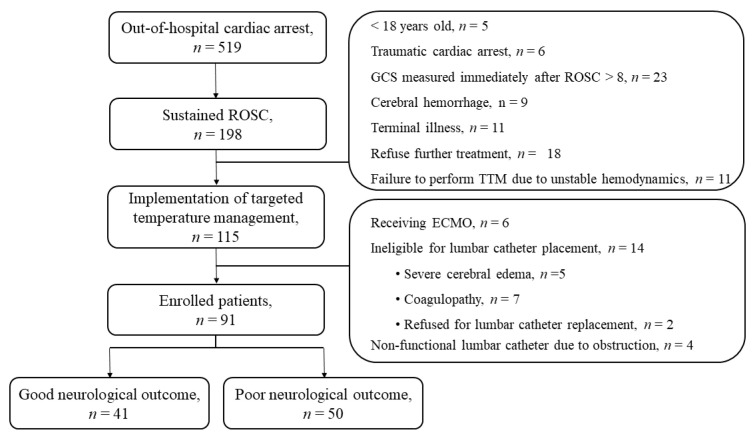
Flow diagram of patient inclusion. ECMO: extracorporeal membrane oxygenation; GCS: Glasgow coma scale; ROSC: return of spontaneous circulation; TTM: targeted temperature management.

**Figure 3 jcm-10-05697-f003:**
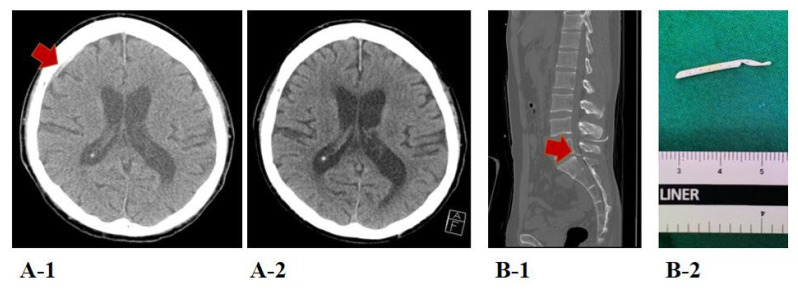
Adverse events during lumbar catheter placement. A 72-year-old female patient was found to have a new subdural hematoma in the right frontal area (red arrow) on a brain computed tomography (CT) scan (**A-1**) taken after targeted temperature management (TTM), which had disappeared on a follow-up brain CT scan (**A-2**) taken 2 weeks later. A 38-year-old male patient had a new residual lumbar catheter tip (red arrow) at the level of S1 on lumbar CT scan (**B-1**) taken before TTM, which was surgically removed 12 days later (**B-2**).

**Figure 4 jcm-10-05697-f004:**
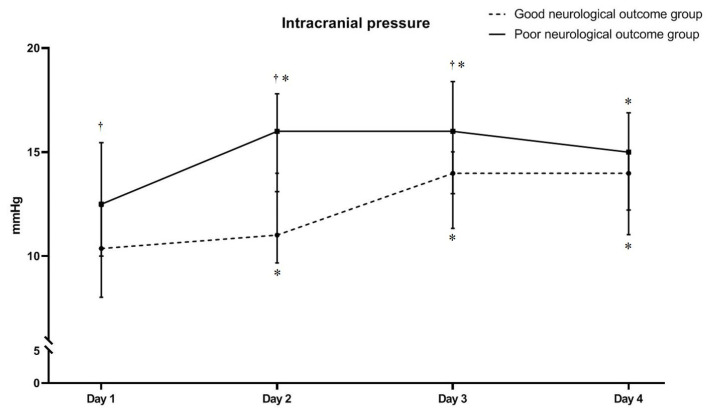
A comparison of intracranial pressure for different neurological outcomes at each time point. The poor neurological outcome group had significantly higher intracranial pressure values (median with interquartile range) than the good neurological outcome group, except on day 4. Furthermore, the degree of elevation in ICP values differed between the two groups, with the highest degree of elevation observed between days 2 and 3 in the good neurological outcome group and between days 1 and 2 in the poor neurological outcome group. * *p*-value < 0.008 (alpha, 0.05/6), pairwise multiple comparison between the first ICP levels and others using a Friedman test with a Wilcoxon-signed rank test for Bonferroni correction. ^†^ *p*-value < 0.05, a Mann–Whitney *U* test was performed to compare the ICP levels between the neurological outcome groups for each time point.

**Figure 5 jcm-10-05697-f005:**
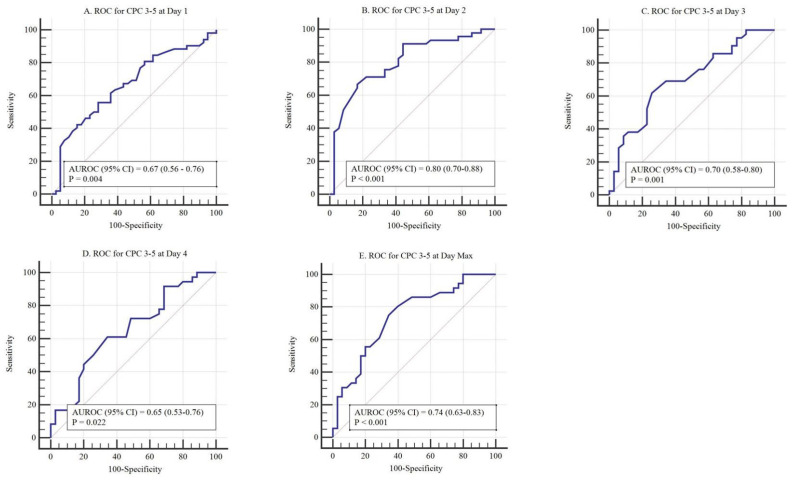
Area under the curves (AUC) for predicting a poor neurological outcome at different intracranial pressure (ICP) levels. (**A**) Receiver operating characteristic curve for ICP levels on day 1 (AUC 0.67, *p* = 0.004; cut-off value > 11.8 mmHg). (**B**) Receiver operating characteristic curve for ICP levels on day 2 (AUC 0.80, *p* < 0.001; cut-off value > 14.0 mmHg). (**C**) Receiver operating characteristic curve for ICP levels on day 3 (AUC 0.70, *p* = 0.001; cut-off value > 15.0 mmHg). (**D**) Receiver operating characteristic curve for ICP levels on day 4 (AUC 0.65, *p* = 0.022; cut-off value > 14.8 mmHg). (**E**) Receiver operating characteristic curve for ICP levels on day max (AUC 0.74, *p* < 0.001; cut-off value > 15.0 mmHg).

**Table 1 jcm-10-05697-t001:** Baseline demographic data and arrest characteristics.

Characteristics	Cohort (*n* = 91)	Good Neurological Outcome (*n* = 40)	Poor Neurological Outcome (*n* = 51)	*p*-Value
Age, years, median (IQR)	57.0 (41.0–69.0)	57.0 (39.5–68.0)	57.0 (42.0–72.0)	0.384
Male sex, *n* (%)	67 (73.6)	32 (80.0)	35 (68.6)	0.242
Charlson Comorbidity Index score, median (IQR)	1.0 (0.0–5.0)	2.0 (0.0–5.0)	1.0 (0.0–5.0)	0.609
Arrest characteristics				
Witness, *n* (%)	62 (68.1)	32 (80)	30 (58.8)	0.032
Location of arrest, public place, *n* (%)	27 (29.7)	13 (14.3)	14 (15.4)	0.256
Bystander CPR, *n* (%)	63 (69.2)	33 (82.5)	30 (58.8)	0.016
Shockable rhythm, *n* (%)	26 (28.6)	22 (55.0)	4 (7.8)	<0.001
Cardiac etiology, *n* (%)	36 (39.6)	24 (60.0)	12 (23.5)	0.001
No flow time, min, median (IQR)	2.0 (0–12.5)	1.0 (0.0–5.0)	6.5 (0.0–23.3)	0.003
Low flow time, min, median (IQR)	19.0 (9.0–29.5)	14.0 (8.0–19.0)	27.5 (15.0–43.0)	<0.001
Laboratory parameters immediately after ROSC				
pH, median (IQR)	7.14 (7.00–7.32)	7.14 (6.97–7.30)	7.15 (7.00–7.30)	0.542
Lactate, mmol l^−1^, median (IQR)	6.20 (3.60–10.80)	4.50 (1.70–10.40)	7.30 (4.08–11.00)	0.038
Albumin, g dl^−1^, median (IQR)	3.2 (2.9–3.6)	3.4 (2.7–3.7)	3.1 (2.7–3.6)	0.005
Creatinine, mg dl^−1^, median (IQR)	1.35 (1.02–7.11)	1.56 (1.12–7.47)	1.25 (0.96–2.65)	0.786
Troponin I, ng ml^−1^, median (IQR)	0.66 (0.06–39.9)	0.27 (0.03–30.4)	5.53 (0.11–82.90)	0.037
Procalcitonin, ng ml^−1^, median (IQR)	0.26 (0.05–0.66)	0.19 (0.05–0.60)	0.26 (0.05–2.01)	0.092
SOFA score	10.0 (8.0–12.0)	9.0 (7.5–11.0)	11.0 (9.0–12.0)	0.018
Time-weighted average 24 h from ROSC, mmHg				
PaO_2_	142.2 (124.7–158.3)	140.9 (116.2–157.3)	142.9 (129.1–162.2)	0.111
PaCO_2_	39.2 (35.4–42.7)	41.0 (36.7–43.5)	38.2 (33.8–42.0)	0.06
MAP	94.0 (86.6–99.9)	98.5 (94.2–101.5)	90.5 (82.1–95.9)	<0.001
Time-weighted average 72 h from ROSC				
NMBA (Nimbex^®^), μg kg^−1^ min^−1^	3.00 (3.00–3.97)	3.00 (3.00–4.00)	3.00 (3.00–3.00)	0.032
Sedative (Midazolam^®^), mg kg^−1^ h^−1^	0.10 (0.06–0.15)	0.10 (0.06–0.12)	0.10 (0.05–0.19)	0.666
ROSC to LP time, hour (IQR)	4.5 (3.2–6.0)	4.1 (3.0–5.9)	4.7 (4.0–6.0)	0.150
ROSC to induction time (33 °C), hour (IQR)	6.0 (4.7–7.5)	5.9 (4.6–8.1)	5.9 (4.8–7.4)	0.962

CPR: cardiopulmonary resuscitation; IQR: interquartile range; LP: lumbar puncture; MAP: mean arterial pressure; PaCO_2_: partial pressure of carbon dioxide; PaO_2_: partial pressure of oxygen; ROSC: return of spontaneous circulation; SOFA: sequential organ failure assessment.

**Table 2 jcm-10-05697-t002:** A comparison of daily intracranial pressure changes between good and poor neurological outcome groups.

Group	*P* _GEE_	Post Hoc Analysis		
		Bonferroni		
		Day 1–2	Day 2–3	Day 3–4
Good neurological outcome	<0.001	0.282	0.001	1.000
Poor neurological outcome	<0.001	0.001	1.000	1.000

**Table 3 jcm-10-05697-t003:** A comparison of Q_A_ values between neurological outcome groups.

Time	Cohort	Neurological Outcome at 3 Months after ROSC		*p*-Value
		Good	Poor	
Day 1	0.008 (0.006–0.014), 91 *	0.007 (0.005–0.009), 40 *	0.01 (0.007–0.018), 51 *	0.001 **
Day 2	0.018 (0.007–0.054), 81 *	0.008 (0.005–0.017), 36 *	0.03 (0.015–0.137), 45 *	<0.001 **
Day 3	0.015 (0.007–0.043), 78 *	0.007 (0.006–0.013), 35 *	0.029 (0.015–0.096), 43 *	<0.001 **
Day 4	0.016 (0.007–0.031), 71 *	0.007 (0.005–0.013), 35 *	0.025 (0.016–0.068), 36 *	<0.001 **

* Number of patients included in the analysis; ** *p*-values are significant at *p* < 0.05.

## Data Availability

Data are available upon reasonable request and with the approval from the CNUH BRAIN research team.

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
