# Peer review of "Intracranial Pressure Patterns and Neurological Outcomes in Out-of-Hospital Cardiac Arrest Survivors after Targeted Temperature Management: A Retrospective Observational Study"

_jcm, 2021, doi:10.3390/jcm10235697_

Round 1
Reviewer 1 Report
Song et. al present an interesting study with measurement of ICP through lumbal puncture in OHCA survivors undergoing TTM.
Overall comments:
They found that the ICP were elevated in all patients treated with TTM after OHCA, but higher in patients with a poor outcome. Though not significantly immediately after ROSC and after 72 hours.
It may not change the overall results, however repeated measure mixed model analysis could have been a more appropriate method. Repeated measure mixed models would be interesting in terms of a significant interaction between the groups, as the authors state a different course of ICP between the groups. And can be done with incomplete date.
Could such analysis be added?
Specific comments:
Study design and population:
How come you report 37 patients here, when the results and consort diagram reports 91 patients?
Reviewer 2 Report
Comments and suggestions for authors.
Thank you for asking me to review this retrospective observational study describing ICP at day 1, 2, 3 and 4 after OCHA, and neurological status after 3 months. This study gives additional information regarding ICP after cardiac arrest. The ICP is within the normal range both in the group with good neurological outcome and in the group with poor neurological outcome. The ROC analysis of optimal cut-off values of ICP for prediction of poor outcome is less clinical relevant due to poor sensitivity and specificity. This should be further discussed. The discussion is a bit long and need to be more concise and focused on the results in this study.
Minor comments:
Introduction:
Page 1, line 40-41: Many centers worldwide do not use ICP in post-CA care, and I suggest reformulating the following statement: “ICP control plays an important role in the improvement of mortality rate and neurological prognosis in post-CA care.” Please, add a reference.
Page 2, line 44-46: This paragraph is a bit unclear. In most CA patients, ICP is not measured, and the mentioned methods are not necessarily used to measure ICP. I suggest to reformulate this.
Materials and methods
37 of the patients included in this study were part of 2 other studies. Please describe whether the rest of the patients were part of other studies or not.
Page 3, line 116: Please state the size of the Lumbar catheter
Page 4, line 126: Qa is probably not familiar for all readers. Please write what it stands for.
Page 4, line 152-153: “Better CA characteristics” is imprecise. I would recommend to reformulate this.
Figure 5: The curves are blurry and difficult to read. It is not possible to find the cut-off values from the figure, and it is difficult to understand the Figure without reading the text. Please add sufficient information in the legends.
Discussion
I suggest making the discussion shorter with more focus on the results from this study. The discussion starts with a description of the findings – an elevation of ICP regardless of neurological prognosis. ICP levels where significantly higher in the poor outcome group. Here, they should also write something about ICP being within the normal range in all groups. A better description of the clinical importance of their findings could also be appropriate in the discussion. They have several references and paragraphs that seems less relevant for this article.
Page 9, line 250-259: The references to Naito et al and Sekhon et al. How do you relate these studies to your results?
Page 9, line 260. Why do the authors think a “reinterpretation of our findings may provide a better understanding of their results”. This should be better explained or deleted.
Page 10, line 272-282: Is this paragraph relevant?
Page 10, line 290: The ICP cut-off levels predicting poor neurological outcomes seems not clinical relevant and have rather low sensitivity and specificity. This needs a comment.
Page 10, line 299-300: This statement about the current consensus is interesting. It needs a reference and perhaps an explanation. I do not think that this is a worldwide consensus.
Conclusion:
I suggest adding that all ICPs where within the normal range. And delete the last sentence (line 331-332)
